# Novel 2,5-Diketopiperazines with In Vitro Activities against Protozoan Parasites of Tropical Diseases

**DOI:** 10.3390/ph17020223

**Published:** 2024-02-08

**Authors:** Isabela P. Ceravolo, Letícia F. Leoni, Antoniana U. Krettli, Silvane M. F. Murta, Daniela de M. Resende, Mariza G. F. de M. L. Cruz, Jodieh O. S. Varejão, Lorena L. Mendes, Eduardo V. V. Varejão, Markus Kohlhoff

**Affiliations:** 1Laboratory of Immunopathology, René Rachou Institute (IRR), Oswaldo Cruz Foundation (FIOCRUZ), Av. Augusto de Lima, 1715, Belo Horizonte 30190-002, Brazil; isabela.ceravolo@fiocruz.br (I.P.C.); leticia.leoni@fiocruz.br (L.F.L.); antoniana.krettli@fiocruz.br (A.U.K.); 2Laboratory of Functional Genomics of Parasites, René Rachou Institute (IRR), Oswaldo Cruz Foundation (FIOCRUZ), Av. Augusto de Lima, 1715, Belo Horizonte 30190-002, Brazil; silvane.murta@fiocruz.br (S.M.F.M.); daniela.resende@fiocruz.br (D.d.M.R.); mcruz@aluno.fiocruz.br (M.G.F.d.M.L.C.); 3Laboratory of Natural Product Chemistry Studies and Organic Synthesis, Federal University of Viçosa (UFV), Av. PH Rolfs, s/n, Viçosa 36570-900, Brazil; jodieh.varejao@ufv.br (J.O.S.V.); lorenalessa02@gmail.com (L.L.M.); eduardo.varejao@ufv.br (E.V.V.V.); 4Laboratory of Bioactive Natural Product Chemistry, René Rachou Institute (IRR), Oswaldo Cruz Foundation (FIOCRUZ), Av. Augusto de Lima, 1715, Belo Horizonte 30190-002, Brazil

**Keywords:** diketopiperazines, cyclic dipeptides, malaria, Chagas disease, leishmaniasis, *Plasmodium falciparum*, *Trypanosoma cruzi*, *Leishmania infantum*

## Abstract

Malaria, Chagas disease, and leishmaniasis are tropical diseases caused by protozoan parasites of the genera *Plasmodium*, *Trypanosoma* and *Leishmania*, respectively. These diseases constitute a major burden on public health in several regions worldwide, mainly affecting low-income populations in economically poor countries. Severe side effects of currently available drug treatments and the emergence of resistant parasites need to be addressed by the development of novel drug candidates. Natural 2,5-Diketopiperazines (2,5-DKPs) constitute N-heterocyclic secondary metabolites with a wide range of biological activities of medicinal interest. Its structural and physicochemical properties make the 2,5-DKP ring a versatile, peptide-like, and stable pharmacophore attractive for synthetic drug design. In the present work, twenty-three novel synthetic 2,5-DKPs, previously synthesized through the versatile Ugi multicomponent reaction, were assayed for their anti-protozoal activities against *P. falciparum*, *T. cruzi*, and *L. infantum*. Some of the 2,5-DKPs have shown promising activities against the target protozoans, with inhibitory concentrations (IC_50_) ranging from 5.4 to 9.5 µg/mL. The most active compounds also show low cytotoxicity (CC_50_), affording selectivity indices ≥ 15. Results allowed for observing a clear relationship between the substitution pattern at the aromatic rings of the 2,5-DKPs and their corresponding anti-*Plasmodium* activity. Finally, calculated drug-like properties of the compounds revealed points for further structure optimization of promising drug candidates.

## 1. Introduction

Malaria and tropical neglected diseases, such as Chagas disease (CD) and leishmaniasis, continue a global public health burden. The World Health Organization (WHO) lists 247 million cases of malaria worldwide, with an estimated number of more than 0.6 million deaths in 2021 [1]. Regarding CD, over 6 million people are infected with the parasite *Trypanosoma cruzi* in Latin American countries [2,3]. Also, more than 1 million cases annually of leishmaniasis have been reported, with 1 billion people living in risk areas. The main current therapies include Artemisinin in the form of combination therapy (ACT) as anti-malaria drug [1]; Benznidazole and Nifurtimox for CD treatment [4]; and pentavalent antimonials, Amphotericin B, Paromomycin, and Miltefosine against leishmaniasis [5]. Unfortunately, these therapies are challenged by variable efficiency, severe side effects, long treatment regimens, as well as the development of resistant parasites to the available drugs. Therefore, the investigation and development of novel drug candidates for the treatment of such diseases is urgent. However, as these diseases occur mainly in regions where populations are severely impacted by great economic poverty and social vulnerability, there is little commercial incentive by pharmaceutical companies.

A promising platform for therapeutic exploration are 2,5-Diketopiperazines (2,5-DKPs). Chemically, these compounds are characterized by a six-membered bis lactam ring whose carbon and nitrogen atoms afford four points for chemical modifications that may allow for obtaining a vast range of derivatives. Their rigid cyclic backbone mimics natural peptide conformations in combination with stability against rapid metabolic degradation by proteolysis and makes the DKP core an attractive lead-like scaffold for drug discovery [6]. Natural 2,5-DKPs with wide structural diversity and biological properties are found as secondary metabolites in plants, fungi, bacteria, and marine organisms [7,8]. The most structurally simple natural 2,5-DKPs are biosynthesized through head-to-tail condensations of two α-amino acids [9] followed by intramolecular cyclization of the dipeptides, as illustrated in Figure 1. Many natural 2,5-DKPs have shown a broad spectrum of medicinal activities including anticancer, neuroprotective, antioxidant, and antimicrobial properties [9,10]. Others have also shown promising anti-protozoal activities against leishmaniasis [11,12], CD [13,14], and malaria parasites [15,16].

In the search for new drug prototypes to fight these tropical parasite diseases, the known potential of such natural 2,5-DKPs prompted us to investigate the in vitro anti-protozoan activity of a series of synthetic 2,5-DKPs produced through the Ugi four-component reaction (Ugi-4CR) with subsequent intramolecular N-alkylation of the generated Ugi-adducts (Table 1) [17,18]. Ugi-4CR is a versatile, one-step multicomponent reaction that involves the one-pot reaction of an aldehyde, an amine, an acid, and an isocyanide to provide an α-N-acylamino amide (Ugi adduct). Depending on the structural features of the acid or isocyanide, the Ugi adduct can further undergo an intramolecular cyclization to afford the 2,5-DKP ring. The detailed methodology employed for the synthesis of the 2,5-DKPs investigated in the present work, as well as the spectrometric and spectroscopic data are extensively presented and discussed in Mendes et al. [18]. Results from in vitro tests of these 2,5-DKPs against *P. falciparum*, *T. cruzi* and *L. infantum* and the influence of structural features on their activity are presented and discussed.

## 2. Results and Discussion

In initial screenings, the twenty-three 2,5-DKPs were tested at 20 µg/mL for their anti-protozoal activity, and those that inhibited at least 70% of the growth of *P. falciparum* and *T. cruzi* and 60% the growth of *L. infantum* were subjected to further assays to determine their half-maximal inhibitory concentrations (IC_50_) against the target parasites. The obtained results are presented in Table 2. Compounds that exhibited IC_50_ values ≤ 10 µg/mL against the targeted parasites were considered active, those with 20 µg/mL > IC_50_ > 10 µg/mL as partially active, and those with IC_50_ ≥ 20 µg/mL as inactive. The active and partially active compounds were then tested against human and monkey cell lines to assess their potential cytotoxicity. Their half-maximum cytotoxic concentrations (CC_50_) and their selectivity indices (SI), obtained as the ratio of the cytotoxic concentration against the bioactive concentration are shown in Table 2. Theoretically, the higher the SI value, the greater the effectiveness for the target and the lower the toxicity of the compound, characterizing a potentially safe and selective drug [19].

As shown in Table 2, of seven compounds that showed activity against *P. falciparum* in the initial screening, two (17 and 19) have proven to be active (IC_50_ < 10 µg/mL). These two compounds have shown low cytotoxicity resulting in a high selectivity index (SI > 10) and thus being considered as promising lead structures. These results allow for inferring about the influence of the nature and position of the substituents at the aromatic rings on the activity of the compounds against *P. falciparum*. Both compounds bear a fluorine atom at the ortho position of ring B (substituent R_2_ from the structure of the aldehyde, Figure 2), a chlorine atom at the para position of ring A (substituent R_1_ from the isonitrile), and another halogen atom (Cl or Br) at the para position of ring C (R_3_ from aniline derivative). Like compounds 17 and 19, the 2,5-DKPs 5 and 20 also show a fluorine atom at the ortho position of ring B. However, compound 5 shows no substituent at the para position of ring C, while compound 20 has a fluorine atom at this position. Both compounds, 5 and 20, were only partially active though toxic, which implies that not only the presence but also the nature of the halogen atom at this position is important for the biological activity of these 2,5-DKPs. These findings show that the fluorine atom at the ortho position of ring B (which comes from the aldehyde used as reactant) is the main factor for the activity of such compounds against *P. falciparum*.

Anti-malarial or anti-plasmodial activities have been described for a series of 2,5-DKPs. Gancidin W, a cyclo(Pro-Leu) dipeptide was isolated as the main compound from an endophytic *Streptomyces* strain obtained from a Malaysian medicinal plant and tested in vivo as an active, low-toxic 2,5-DKP [20]. Naseseazine C, a structurally more complex dimeric DKP containing an indol ring was isolated from a marine sediment derived *Streptomyces* species and demonstrated an anti-plasmodial activity with IC_50_ = 3.5 µM [21]. Nicolaou et al. [22] have synthesized a series of epidithio- and bis(methylthio)-diketopiperazine derivatives based on the naturally occurring Epicoccin G. Some of the sulfur-containing compounds exhibited IC_50_ against *P. falciparum* in the low micromolar range. Another sulfur-containing 2,5-DKP, 1-demethylhyalodendrin tetrasulfide, isolated from the entomopathogenic fungus *Verticillium hemipterigenum* showed an IC_50_ of 6.7 µM, but high cytotoxicity when tested against human and monkey cell lines [23]. Also, a series of ten simple 2,5-DKPs, synthesized by condensation and cyclization of amino acids, were assayed against *P. berghei* and exhibited inhibitory concentrations ranging from 2.3 to 123.0 µM [16]. Although many 2,5-DKPs show promising inhibition capacities the lack of cytotoxicity tests often hampers the estimation of their selectivity for the parasite and their suitability as potential drug leads. A promising novel 2,5-DKP drug prototype is Diatretol, an α,α′-dioxo-diketopiperazine originally isolated from the fungus *Clitocybe diatrea* [24] and recently synthesized by Takahashi et al. [25]. It demonstrated potent selective plasmodicidal in vitro activity with IC_50_ = 0.6 µM and SI > 250 as well as growth inhibition of the parasite in vivo of more than 50% after application of 30mg/(kg × day) conducting a 4-day suppressive test [15,26]. A substantial activity breakdown of each of the non-oxidized cyclic dipeptide DL-isomers indicates a crucial role of the oxo groups for its activity.

In our present work, the most active 2,5-DKP against P. falciparum (compound **19**) showed an IC_50_ of 5.4 µg/mL, which corresponds to 12.6 µM. Compared to Diatretol, it is only moderately active. However, it is worth mentioning that the 2,5-DKPs assayed in the present work have a quite simple structure and can be readily obtained through the Ugi four component reaction (Ugi-4CR) [17]. This synthesis route consists of a single step, one-pot condensation of an aldehyde, an amine, an acid, and an isocyanide to provide an α-N¬-acylamino amide (Ugi adduct) that can undergo a further cyclization to produce the 2,5-DKP core. By varying the structures of any of the four reactants, the Ugi reaction allows the generation of a large range of structurally diverse 2,5-DKPs. The reaction takes place through a minimum consumption of solvents and the 2,5-DKPs are obtained as solids that precipitate from the reaction medium and can be isolated by simple filtration. Despite the moderate activity of the tested 2,5-DKPs against *P. falciparum*, considering the versatility and low cost of such reactions, these structures may constitute a promising starting point for the development of novel, highly active, and safe anti-plasmodial drugs.

From the results against *T. cruzi*, unfortunately, no rational influence of the substitution pattern at the aromatic rings on the activity of the compounds could be found. However, compounds 8 and 22 have shown good anti-trypanosomal activities with high selectivity indices of SI > 50. The same was observed for 14 against *L. infantum*. For these protozoans, the synthesis of new 2,5-DKPs with greater variability in the substituent groups may bring more information about the relationship between structure and activity toward the development of more potent molecules. Only a few anti-*Trypanosoma* 2,5-Diketopiperazins have been described in the literature so far. Active natural products found in extracts of bacterial symbionts of marine sponges were considered to be related to the putatively annotated dipeptides cyclo(Ile-Pro) and cyclo(Phe-Pro) [27]. In a more detailed study [14], natural and semi-synthetic DKPs from marine-derived fungi were screened and revealed some bioactive compounds. Although diverse in structure most of them were cytotoxic, with only four of them slightly passing the selectivity criteria (SI > 10). In the work reported by Martins-Teixeira et al. [13], two glycosyl diketopiperazines were synthesized and tested for their in vitro activity against *T. cruzi*, showing only weak activities with IC_50_ > 100 µM.

In assays against *L. infantum*, the two (partially) active compounds (7 and 14) have shown high selectivity indices of SI > 40 but no clear relationship between the patterns of substitution and biological activity. Guimarães Tunes et al. [12] isolated two cyclic dipeptides from the endophytic fungus *Penicillium citrinum* obtained from leaves of *Ageratum myriadena* which have shown weak leishmanicidal activities of IC_50_ > 100 µM. In the work of Maity et al. [11], a series of twelve diketopiperazine synthesized by one-pot self-condensation of substituted α-chlorophenyl acetamides were tested against *L. donovani*. Although those 2,5-DKPs also own aromatic rings with halogen atoms as substituents, they differ from the 2,5-DKPs tested in the present work, as they show identical substitution patterns of the N-linked moieties and lack residues at the carbon atoms of the 2,5-DKP core. Most of those 2,5-DKPs showed high activities against *L. donovani*, with IC_50_ < 10 µM on extra cellular promastigotes. Due to their limited structural variability, no clear structure-activity relationship can be inferred, but the results suggest that they show potential for the development of novel anti-leishmanial drugs. Even the 2,5-DKPs regarded as active in the present work have shown only moderate activities compared to those. On the other hand, the low cytotoxicity for these compounds encourages the synthesis and biological evaluation of other structurally diverse 2,5-DKPs that may provide further anti-leishmanial compounds and allow for a deeper elucidation of their structure-activity relationship.

Although the active DKPs described in this study exhibit already moderate anti-protozoal activities together with promising selectivity indices, in vivo tests and further drug optimization are needed to improve their efficiency and bioavailability. Active candidates that exceed a selectivity index of 10 might be tested in vivo to elucidate and monitor their action and compatibility within a mammal model organism. To evaluate the drug- and lead-like properties of the compounds and to check points for modifications, the molecule characteristics for absorption, distribution, metabolism, and excretion (ADME) were predicted using the SwissADME web resource [28].

Table 3 summarizes parameters as the “Rules of 5” [29], structure rigidity, polarity, water solubility, absorption, as well as drug- and lead-likeness. All active DKPs are predicted to be moderately to poorly water soluble but to have good gastrointestinal absorption. They fulfill most of the “rules of 5” and the requirements of a limited polar surface area (TPSA) and structure stiffness, which is expressed as number of rotating bonds (NRB). Only 17, 19, and 22 are too lipophilic to be drug-like. To evaluate a compound to be a potential pharmacophore for further drug development, the lead-likeness restricts the prediction rules to a smaller (<350 Da), more rigid (NRB < 7), and less lipophilic (XLOGP < 3.5) core structure [30]. Due to their three phenyl moieties, all tested DKPs are too large and lipophilic to serve as leads in their current form. As an encouraging example for both good drug- and lead-likeness, the anti-malarial drug candidate Diatretol is added to the list.

Figure 3 gives a rapid visual impression of the likeness as an oral drug in the form of a bioavailability radar and demonstrates Diatretol as a good candidate with its radar points within the red area. Since all the active DKPs in this study are derivatives of the same tri-Phenyl-DKP core, they show a similar radar as 18, clearly ultra-passing the insaturation limit, which requires a fraction of at least one quarter sp3-carbons. Thus, a first step to adapt and improve the synthetic DKPs as drug candidates against parasites causing human diseases might be a partial elimination and rearrangement of the phenyl moieties. Finally, 2,5-Diketopiperazines and cyclic dipeptides are still underexplored as anti-protozoal compounds and exhibit great potential as future drug alternatives against these tropical diseases.

## 3. Materials and Methods

### 3.1. 2,5-Diketopiperazines

Compounds **1**–**23** were synthesized and characterized as published in Mendes et al. [18]. Briefly, a series of substituted aromatic isocyanides, benzaldehydes, and aniline derivatives were reacted with chloroacetic acid through Ugi four-component reactions to produce a series of the corresponding Ugi adducts [17]. Such adducts were subsequently subjected to cyclization through intramolecular N-alkylation to produce the corresponding 2,5-DKPs. As depicted in the scheme of Table 1, 2,5-Diketopiperazines were obtained from substituted aromatic isocyanides, benzaldehydes, aniline derivatives, and chloroacetic acid through intramolecular N-alkylation of Ugi adducts. The chemical structures of all 2,5-DKPs were undoubtedly confirmed by spectrometric, spectroscopic, and crystallography experiments. All data (FT-IR-ATR, 1 D and 2 D NMR, IEMS, HRMS, and X-ray crystallographic data) are presented and discussed in [18]. For activity tests in bioassays, the compounds were dissolved in DMSO at concentrations of 20 mg/mL, and their identity and purity were confirmed by UHPLC-HRMS.

### 3.2. UHPLC-HRMS

Analyses were performed on a Nexera UHPLC-system (Shimadzu, Kyoto, Japan) hyphenated to a maXis high-resolution ESI-QTOF mass spectrometer (Bruker, Bremen, Germany) controlled by the Compass 1.7 software package (Bruker). 5 μg compound were injected onto a Shimadzu Shim-Pack XR-ODS-III column (C18, 2.2 µm, 2.0 × 150 mm) at 40 °C under a flow rate of 400 μL/min using 0.1% formic acid in both water milliQ and acetonitrile as mobile phases A and B, respectively. Elution started with initial 0.5 min 5% B followed by a linear gradient to 100% B in 10 min and a hold at 100% B for 1.5 min. After UV-PDA-detection (190–450 nm), mass spectra were acquired in positive mode at a spectra rate of 5 Hz. Ion-source parameters were set to 500 V end plate offset, 4500 V capillary voltage, 3.0 bar nebulizer pressure, and a dry gas flow of 8 L/min at 200 °C. Data-dependent fragment spectra were recorded using a collision energy range between 15 and 60 eV. Mass calibration was achieved by initial ion-source infusion of 20 µL calibrant solution (1 mM sodium formiate in 50% 2-propanol) and post-acquisition recalibration of the raw data. Compound confirmation was achieved by chromatographic peak dissection with subsequent formula determination according to exact mass and isotope pattern as well as matching fragment spectra.

### 3.3. Anti-Plasmodium falciparum Activity Assay

In vitro assay with *P. falciparum* culture were performed as described previously [31]. Briefly, the compounds were tested against the parasite erythrocytic asexual stages using a chloroquine-resistant and mefloquine-sensitive W2 clone [32] cultured at 37 °C [33]. The activity was measured using the SYBR assay with the parasite suspension (0.5% parasitemia and 2% hematocrit) [34] with minor modifications [31]. The test compounds, in serial dilutions, were incubated in “U” bottom 96-wells plates. After 48 h at 37 °C, the culture supernatant was removed and replaced by 100 µL of lysis buffer solution [Tris (20 mM; pH 7.5), EDTA (5 mM), saponin (0.008%; wt/vol), and Triton X 100 (0.08%; vol/vol)] followed by addition of 0.2 µL/mL SYBR Safe (Sigma-Aldrich, Carlsbad, CA, USA). The plate content was transferred to a flat bottom plate and incubated in the dark for 30 min. The plate was read in a fluorometer (Synergy H4 Hibrid Reader, Biotek) with excitation at 485 nm and 535 nm of emission. In all tests, the compound activities were expressed by the 50% inhibitory concentration of the parasite growth (IC_50_) when compared to the drug-free controls and estimated using the curve-fitting software Origin 8.0 (OriginLab Corporation, Northampton, MA, USA). Compounds with IC_50_ greater than 20 µg/mL were classified as not active, as partially active when 10 µg/mL < IC_50_ < 20 µg/mL, and less than 10 µg/mL were considered as active [35,36]. Chloroquine was used as an anti-malarial reference drug in all tests performed.

For the cytotoxicity assays the monkey kidney cell line (BGM) (ATCC, Manassas, VA, USA) was used and maintained as suggested by the manufacturers. The cells were cultured in 75 cm^2^ bottles with RPMI 1640 medium supplemented with 10% heat-inactivated fetal calf serum and 40 mg/L gentamicin in a 5% CO2 atmosphere at 37 °C. For the in vitro tests, a confluent cell monolayer was treated with trypsin and the cells distributed in a flat-bottomed 96-well plate (3 × 105 cells/mL) and incubated for 18 h at 37 °C to ensure cell adherence. The BGM cells were incubated with 20 µL of the drugs at different concentrations (≤1000 µg/mL) for 24 h in a 5% CO_2_ at 37 °C. The neutral red uptake assay was used to evaluate the lysosomal integrity and distinguishes live cells from dead by its ability to incorporate the dye [37]. Briefly, to each well was added 0.2 mL medium containing 50 µg/mL of neutral red solution. The plate was incubated for another 3 h at 37 °C to allow for the uptake of the vital dye into the lysosomes of viable uninjured cells. After removal of medium, 200 µL of a mixture of 0.5% formaldehyde-1% CaCl2 was added to the cells and incubated by 5 min. The supernatant was removed and 100 µL of a solution of 1% acetic acid-50% ethanol was added to each well to extract the dye. After homogenization, the optical density of each well of the plate was measured using a 540 nm wavelength on a spectrophotometer. This absorbance has shown a linear relationship with the number of surviving cells. The cell viability was expressed as the percentage of control absorbance obtained from the untreated cells after subtracting the appropriate background. The cytotoxic concentration was determined using at least duplicate tests to calculate the cytotoxic dose for 50% of the cells (MLD_50_ or CC_50_), as described [38].

The ratio between drug cytotoxicity (CC_50_) and activity (IC_50_) was used to estimate the selectivity index (SI), as shown before [39], where SI less than 10 was indicative of toxicity.

### 3.4. Anti-Trypanosoma cruzi Activity Assay

The in vitro anti-*T. cruzi* activity was evaluated on L929 cells (mouse fibroblasts) infected with Tulahuen strain of the parasite expressing the *Escherichia coli* β-galactosidase as reporter gene according to the method described previously [40,41]. Briefly, four thousand L929 cells were added to each well of a 96-well microtiter plate. After an overnight incubation, a 10 fold excess of trypomastigotes were added to the cells, incubated for 2 h, then replaced with 200 μL of fresh medium and incubated for additional 48 h to establish the infection. For IC_50_ determination, the cells were exposed to each synthesized compound at serial decreasing dilutions and the plate was incubated for 96 h. Afterwards, 50 μL of 500 μM chlorophenol red beta-D-galactopyranoside (CPRG) in 0.5% Nonidet P40 was added to each well, and the plate incubated for 16 to 20 h, after which the absorbance at 570 nm was measured. Controls with uninfected cells, untreated infected cells, infected cells treated with benznidazole at 3.8 μM (positive control) or 1% DMSO were used. The results were expressed as the percentage of *T. cruzi* growth inhibition in compound tested cells as compared to the infected cells and untreated cells. The IC_50_ values were calculated by linear interpolation. Quadruplicates were run in the same plate, and the experiments were repeated at least once.

For the in vitro cytotoxic activity test and CC_50_ determination over the L929 cell line, 4000 L929 cells in 200 μL of RPMI-1640 medium (pH 7.2–7.4) (Gibco BRL) plus 10% fetal bovine serum and 2 mM glutamine were added to each well of a 96-well microtiter plate that was incubated for three days at 37 °C. The medium was then replaced, and the cells were exposed to compounds at increasing concentrations starting at IC_50_ value for *T. cruzi*. After 96 h of incubation with the compounds, AlamarBlue^TM^ was added and the absorbance at 570 and 600 nm was measured after 4–6 h. Controls with untreated and DMSO 1%-treated cells were run in parallel. The results were expressed as the percent difference in the reduction between treated and untreated cells. The compound concentration that inhibits 50% of the L929 cell viability (CC_50_) was determined. Quadruplicates were run in the same plate and the experiments were repeated at least once.

IC_50_ over *T. cruzi* and L929 cells were determined by linear interpolation and the selectivity index (SI) was calculated by the ratio of CC_50_ L929 cells/ IC_50_
*T. cruzi*.

### 3.5. Anti-Leishmania Infantum Activity Assay

Cells derived from the human monocytic strain THP-1 were cultured in complete Rowell Park Memorial Institute (RPMI)-1640 medium (supplemented with 10% fetal bovine serum, 100 U/mL penicillin, and 100 μg/mL streptomycin). Monocytes were differentiated into macrophages by the addition of 20 ng/mL phorbol myristate acetate (PMA). After 72 h, the macrophages (5 × 104), in black 96-well microtiter plates (Corning Incorporated, Corning, NY), were infected with promastigotes forms of *Leishmania* (*Leishmania infantum* (MHOM/BR/1974/PP75) expressing red fluorescence protein [42], on the second day of the stationary phase (20 parasites per macrophage) for 4 h. The parasites that failed to infect the macrophages were washed away (three times with 1× PBS), and the infected macrophages were incubated for 72 h in RPMI-1640 medium containing different compounds concentrations (80, 40, 20, 10 and 5 μg/mL). The compound concentration that inhibits 50% of parasite growth (IC_50_) was determined by the decrease in parasite fluorescence in the absence and presence of the compound, after 72 h of exposure, using a microplate reader (SpectraMax M2, Molecular Devices, Sunnyvale, CA, USA) with excitation and emission values: 554 and 581 nm, respectively. The IC_50_ values were determined by non-linear regression using a variable slope model (log[concentration] vs. growth inhibition) with the GraphPad Prism v8.2.0 (Boston, MA, USA) software. Controls with uninfected cells, untreated infected cells, infected cells treated with amphotericin B at 0.25 μM (positive control) or DMSO 1% were used. Quadruplicates were run in the same plate, and the experiments were repeated at least once.

For the cytotoxicity assay, human monocytic THP-1 cells were differentiated into macrophages in 96-well microtiter plates for 72 h, as described above. The medium was then replaced, and the cells were exposed to the compounds at increasing concentrations starting at IC_50_ value for each *Leishmania* spp. line. After 72 h of incubation with the compounds, AlamarBlue™ was added and the absorbance at 570 and 600 nm was measured after 4 h. Controls with untreated and DMSO 1%-treated cells were run in parallel. The results were expressed as the percentual difference in the reduction between treated and untreated cells. The compound concentration that inhibits 50% of the THP-1 cell viability (CC_50_) was determined. The CC_50_ values were determined by non-linear regression using a variable slope model (log(concentration) vs. growth inhibition) with the GraphPad Prism v8.2.0 (CA, USA) software. Triplicates were run in the same plate and the experiments were repeated at least once.

## Figures and Tables

**Figure 1 pharmaceuticals-17-00223-f001:**
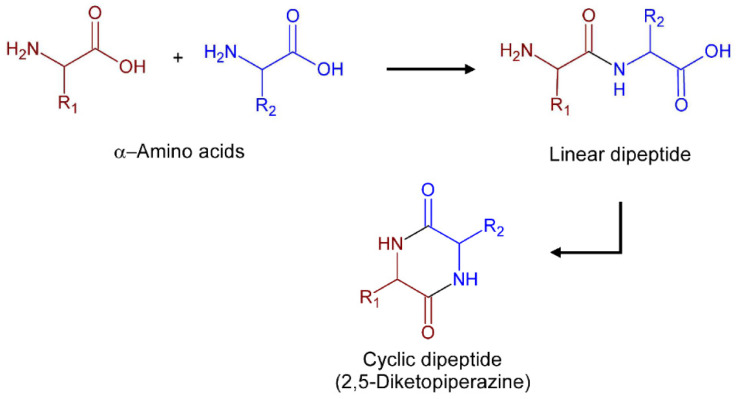
Schematic representation of the formation of cyclic dipeptides from amino acids.

**Figure 2 pharmaceuticals-17-00223-f002:**
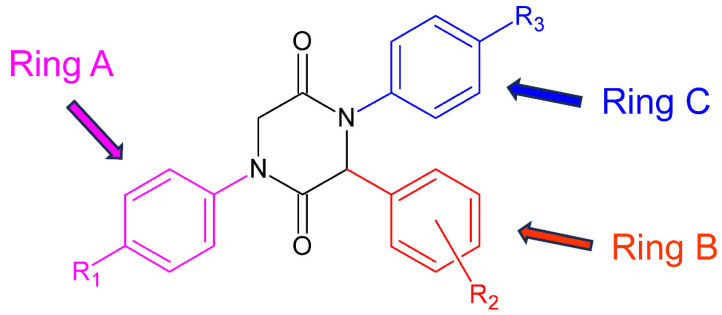
General structure and ring notation of the synthesized 2,5-DKPs.

**Figure 3 pharmaceuticals-17-00223-f003:**
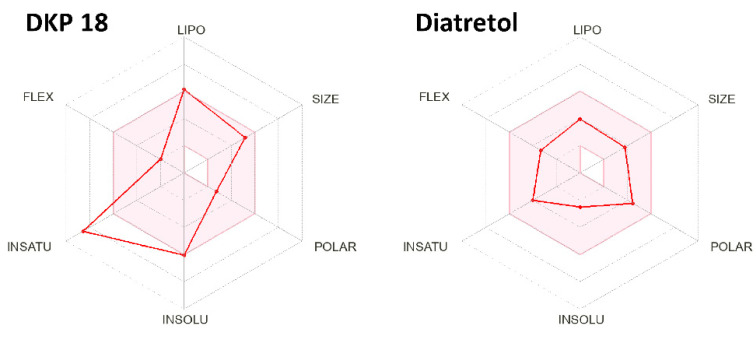
Bioavailability radar examples of the active 2,5-DKPs **18** and Diatretol. The red area represents the optimal range of LIPOphilicity, SIZE, POLARity, INSOLUbility, INSATUration, and FLEXibility for drug-like compounds.

**Table 1 pharmaceuticals-17-00223-t001:** Synthetic 2,5-Diketopiperazines tested for anti-protozoal activity.

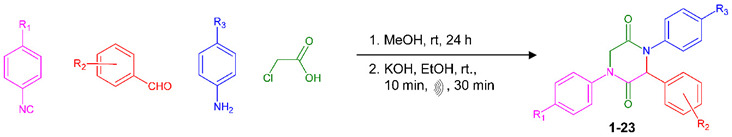
Compound	R_1_	R_2_	R_3_
**1**	Cl	H	H
**2**	Cl	4-Et	H
**3**	Cl	4-NMe_2_	H
**4**	Cl	4-Cl	H
**5**	Cl	2-F	H
**6**	Cl	4-OMe	H
**7**	Cl	4-NHCOMe	H
**8**	Cl	3-Br, 4-OMe	H
**9**	Cl	2,5-OMe	H
**10**	Cl	3-Cl	H
**11**	Cl	4-F	H
**12**	OMe	4-OMe	OMe
**13**	OMe	H	OMe
**14**	Cl	4-OH	H
**15**	Cl	3-Br, 4-OH	H
**16**	Cl	2-F	OMe
**17**	Cl	2-F	Cl
**18**	Cl	4-Cl	OMe
**19**	Cl	2-F	Br
**20**	Cl	2-F	F
**21**	Cl	4-Cl	F
**22**	Cl	2-Cl, 6-F	H
**23**	Cl	2,4,6-OMe	H

**Table 2 pharmaceuticals-17-00223-t002:** Inhibitory concentrations (IC_50_), cytotoxic concentrations (CC_50_), selectivity indices (SI), and activity status of synthetic 2,5-Diketopiperazines tested against tropical protozoan parasites.

Parasite	Compound	CC_50_ (µg/mL) ^a^	IC_50_ (µg/mL) ^b^	SI ^c^	Status
*P. falciparum*	**5**	≤3.1	15.6 ± 2.6	<1	toxic
	**10**	NT	28.1 ± 0.3	NA	inactive
	**15**	18.5 ± 3.4	14.9 ± 0.7	1	toxic
	**17**	138.6 ± 18.7	9.5 ± 2.1	15	ACTIVE
	**19**	110.0 ± 11.3	5.4 ± 1.7	20	ACTIVE
	**20**	9.4 ± 5.1	17.6 ± 1.3	<1	toxic
	**21**	NT	≥50	NA	inactive
	Chloroquine	221.8 ± 28.8	0.174 ± 0.055	1257	Control
*T. cruzi*	**5**	29.2	9.3 ± 0.1	3	toxic
	**8**	>400	7.3 ± 1.5	>55	ACTIVE
	**13**	NT	22.3 ± 1.1	NA	inactive
	**15**	81.8	11.2 ± 6.0	7	toxic
	**20**	32.1	15.4 ± 3.0	2	toxic
	**21**	16.8	16.1 ± 2.0	1	toxic
	**22**	>400	7.2 ± 0.9	>56	ACTIVE
	Benznidazole	540	0.5 ± 0.1	1080	Control
*L. infantum*	**2**	NT	>100	NA	inactive
	**3**	NT	49.1 ± 1.6	NA	inactive
	**4**	NT	35.7 ± 0.8	NA	inactive
	**7**	679 ± 1	16.0 ± 1.0	42	part. ACTIVE
	**9**	NT	50.8 ± 1.1	NA	inactive
	**11**	NT	38.2 ± 1.8	NA	inactive
	**14**	611 ± 1	8.3 ± 0.2	74	ACTIVE
	Amphotericin B	367 ± 1	0.228 ± 0.025	1610	Control

^a^ Cytotoxic concentration for 50% cells; ^b^ concentration for 50% inhibition of parasite cell infection, IC_50_ ≥ 20 µg/mL inactive, 20 > IC_50_ > 10 µg/mL partially active, ≤10 µg/mL active; ^c^ selectivity index = CC_50_/IC_50_, values < 10 are considered toxic; NT = not tested; NA = not applicable.

**Table 3 pharmaceuticals-17-00223-t003:** Drugability properties of the active 2,5-DKPs and Diatretol.

Compound	MW (Da)	HBA	HBD	Log P_(o/w)_	NRB	TPSA (Å^2^)	Water Solubility	GI	Drug-Likeness *	Lead-Likeness *
Rules	≤500	≤10	≤5	≤4.15	≤9	≤130				
**7**	433.89	3	1	3.01	5	69.72	o	+	0	2
**8**	485.76	3	0	3.86	4	49.85	−	+	0	2
**14**	392.83	3	1	3.07	3	60.85	o	+	0	2
**17**	429.27	3	0	4.48	3	40.62	−	+	1	2
**19**	473.72	3	0	4.58	3	40.62	−	+	1	2
**22**	429.27	3	0	4.48	3	40.62	−	+	1	2
Diatretol	306.36	4	3	0.86	5	87.66	+	+	0	0

MW: Molecular Weight, NRB: Number of rotating bonds, HBA/D: Number of hydrogen bond acceptors/donators, TPSA: Topological polar surface area, P_(o/w)_: Partition coefficient (n-octanol/water, MLOGP), GI: Gastrointestinal absorption. −/o/+: poor/moderately/high. * Numbers of rule violations [29,30]. Values were predicted by SwissADME [28].

## Data Availability

The datasets generated and/or analyzed during the current study are available from the corresponding author on reasonable request.

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
