# Peer review of "Novel 2,5-Diketopiperazines with In Vitro Activities against Protozoan Parasites of Tropical Diseases"

_pharmaceuticals, 2024, doi:10.3390/ph17020223_

Round 1
Reviewer 1 Report
Comments and Suggestions for Authors
The article titled «Novel 2,5-Diketopiperazines with in-vitro activities against Protozoan Parasites of Tropical Diseases» is devoted to synthesis and biological evaluation of 23 new compounds. The article corresponds to modern trends in the search for new pharmacological agents and deserves publication after minor changes.
1) In vitro and in vivo should be written without hyphen and italicized. Please check all text.
2) Lines 74-80, table 1, and the chemical part with a little explanation of reaction, definition of chosen components, yields, and key characteristic of NMR signals should be included into Results and discussion part.
3) The reference No 18 is missed (after 17 (line 78) follow 19 (line 101)). The same with ref 22
4) Line 135 - Nicolaou et al. (2012) must be specified as a number in brackets, the same with Mendes et al. (line 251)
5) Materials and methods should contain all 1H and 13C NMR data of new compounds
6) All IC50 should be provided as underscript
Comments on the Quality of English Language
Minor editing of English language required
Reviewer 2 Report
Comments and Suggestions for Authors
This is an interesting, nicely designed and performed paper which provides some insights into the search for new drugs as alternative resources for the treatment of severe tropical diseases caused by protozoan parasites such as malaria, visceral leishmaniasis and trypanosomiasis. It is particularly foccused on the role played by synthetic cyclic dipeptidase derivatives such as some diketopiperazines (2,5-DKPs).
Results look quite consistent and reliable. Therefore I advise its publication in Pharmaceuticals after minor typing amendments and complementary control assays.
1. All scientific generic and specific names along the manuscript, must be typed in italics (Plasmodium, Trypanosoma, Leishmania, Penicillium citricum Escherichia coli, Streptomyces, and so on....).
2. Due to the high rate of resistance developed by Plasmodium spp., all over the world, perhaps chloroquine may not represent the most reliable control drug. Therefore I suggest to include some other antimalarial compounds such as artemisinin derivative (Artesunate, Artemether....) as complementary or alternative controls.
Comments on the Quality of English Language
I have not remarking comments about English. For me is quite correctly fluent and good level
Reviewer 3 Report
Comments and Suggestions for Authors
According to the World Health Organization (WHO), there are >240 million cases of Malaria worldwide and >0.6 million deaths; over 6 million people are infected by Trypanosoma cruzi, the parasite that causes Chagas Disease, and >1 million cases are annually reported of Leishmaniasis. Malaria, Chagas Disease, and Leishmaniasis are tropical diseases caused by protozoan parasites of the genera Plasmodium, Trypanosoma and Leishmania, respectively. The investigation and development of novel drug candidates for the treatment of such diseases remain urgent. Natural 2,5-Diketopiperazines (2,5-DKPs) constitute N-heterocyclic secondary metabolites with a wide range of biological activities of medicinal interest. The current results suggested that some of the 2,5-DKPs might show promising activities against the protozoan parasites Plasmodium falciparum, Trypanosoma cruzi and Leishmania infantum, with inhibitory concentrations (IC50) ranging from 5.4 to 9.5 μg/ml.
The following comments might improve the work.
TITLE
- Lines 2-3. Since the author have mainly performed experiments on Plasmodium falciparum, Trypanosoma cruzi and Leishmania infantum, the title “Novel 2,5-Diketopiperazines with in-vitro activities against Protozoan Parasites of Tropical Diseases” should be modified accordingly.
INTRODUCTION
- Lines 43-44. Tropical neglected diseases, including Malaria, Chagas disease (CD) and Leishmaniasis, continue a global public health burden.
Do the authors consider Malaria as a neglected tropical disease?
RESULTS AND DISCUSSION
- Lines 93-95. Compounds that exhibited IC50 values ≤ 10 μg/ml against the targeted parasites were considered active, those with 20 μg/ml > IC50 > 10 μg/ml as partially active, and those with IC50 ≥ 20 μg/ml as inactive.
Why do you select 10 μg/ml and 20 μg/ml to classify the compounds? Please, clearly explain the criteria of selection.
- Table 2. You should include the CC50, IC50 and SI values of the reference drugs.
- Why did the authors select the following species (Plasmodium falciparum, Trypanosoma cruzi and Leishmania infantum (visceral leishmaniasis)?
- Did they expect similar results against Cutaneous Leishmaniasis?
CONCLUSION
- A conclusion section or paragraph will be helpful.
